# A strontium isoscape of inland southeastern Australia

**Patrice de Caritat[1], Anthony Dosseto[2], and Florian Dux[2]**

[1]Geoscience Australia, GPO Box 378, Canberra ACT 2601, Australia
[2]Wollongong Isotope Geochronology Laboratory, School of Earth, Atmospheric and Life Sciences,
University of Wollongong, Wollongong NSW 2522, Australia

**Correspondence:** Patrice de Caritat (patrice.decaritat@ga.gov.au)

**Abstract.** The values and distribution patterns of the strontium (Sr) isotope ratio $^{87}Sr/^{86}Sr$ in Earth surface materials are of use in the geological, environmental, and social sciences. Ultimately, the $^{87}Sr/^{86}Sr$ ratios of soils and everything that lives in and on them are inherited from the rocks that are the parent materials of the soil's components. In Australia, there are few large-scale surveys of $^{87}Sr/^{86}Sr$ available, and here we report on a new, low-density dataset using 112 catchment outlet (floodplain) sediment samples covering 529 000 km$^2$ of inland southeastern Australia (South Australia, New South Wales, Victoria). The coarse (<2 mm) fraction of bottom sediment samples (depth ~ 0.6–0.8 m) from the National Geochemical Survey of Australia were milled and fully digested before Sr separation by chromatography and $^{87}Sr/^{86}Sr$ determination by multicollector-inductively coupled plasma mass spectrometry. The results show a wide range of $^{87}Sr/^{86}Sr$ values from a minimum of 0.7089 to a maximum of 0.7511 (range 0.0422). The median $^{87}Sr/^{86}Sr$ ($\pm$ median absolute deviation) is 0.7199 ($\pm$0.0071), and the mean ($\pm$ standard deviation) is 0.7220 ($\pm$0.0106). The spatial patterns of the Sr isoscape observed are described and attributed to various geological sources and processes. Of note are the elevated (radiogenic) values ($\geq\sim$ 0.7270; top quartile) contributed by (1) the Palaeozoic sedimentary country rock and (mostly felsic) igneous intrusions of the Lachlan geological region to the east of the study area; (2) the Palaeoproterozoic metamorphic rocks of the central Broken Hill region; both these sources contribute radiogenic material mainly by fluvial processes; and (3) the Proterozoic to Palaeozoic rocks of the Kanmantoo, Adelaide, Gawler, and Painter geological regions to the west of the area; these sources contribute radiogenic material mainly by aeolian processes. Regions of low $^{87}Sr/^{86}Sr$ ($\leq\sim$ 0.7130; bottom quartile) belong mainly to (1) a few central Murray Basin catchments; (2) some Darling Basin catchments in the northeast; and (3) a few Eromanga geological region-influenced catchments in the northwest of the study area; these sources contribute radiogenic CE1 material mainly by fluvial processes. The new spatial Sr isotope dataset for the DCD (Darling–Curnamona–Delamerianregion) is publicly available (de Caritat et al., 2022; https://dx.doi.org/10.26186/146397).

## 1 Introduction

Strontium (Sr) isotopes in Earth systems are useful to elucidate geoscientific, environmental, and societal issues, including mineralization (e.g. Zhao et al., 2021), hydrology (e.g. Christensen et al., 2018), food tracing (e.g. Vinciguerra et al., 2015), dust provenancing/sourcing (e.g. Revel-Rolland et al., 2006; De Deckker, 2020), and historic migrations of people and animals (e.g. Price et al., 2017; Madgwick et al., 2019, 2021). Whilst large-scale Sr isotopic maps (Sr isoscapes) have been developed for several countries (and even larger areas around the world), recent examples of which include Italy (Lugli et al., 2022), Ireland (Snoeck et al., 2020), Europe (Hoogewerff et al., 2019), and even a world model (Bataille et al., 2020), no national-scale Sr isotope dataset currently exists for Australia. The largest Sr isoscape to-date in this country is for the Cape York Peninsula in northern Queensland (Adams et al., 2019), which covers approximately 300 000 km$^2$. In this contribution, we publish a new Sr isotopic dataset and isoscape for a part of inland south-

eastern Australia across three states, covering agricultural land and remote and arid "outback" deserts. Additional parts of Australia currently are being investigated to expand this isoscape further.

## 2   Setting

The study area, the Darling–Curnamona–Delamerian (DCD) area, is located in inland southeastern Australia across parts of the states of South Australia, New South Wales, and Victoria (hereafter abbreviated to SA, NSW, and Vic, respectively) between 28 and 37° S and 138 and 146° E (Fig. 1). The DCD area is one of the "deep-dive" focus areas targeted for geoscientific data acquisition (https://www.ga.gov.au/eftf/projects/darling-curnamona-delamerian, last access: 22 August 2022) under the Exploring for the Future program (https://www.eftf.ga.gov.au/about TS1, last access: 22 August 2022), a major Australian Government research funding initiative.

The main climate zones in the area are described as "warm summer, cold winter" along the southern/southwestern coastal fringe to "hot dry summer, cold winter" over the remainder of the inland area (BOM, 2021a). The vegetation/Köppen zones are dominated by "grassland/warm (persistently dry)" and "desert/warm (persistently dry)" in the south, and "grassland/hot (persistently dry)" and "desert/hot (persistently dry)" in the north (BOM, 2021b). The 30-year (1976–2005) average minimum temperatures and maximum temperatures mostly range 3–18 and 18–33 °C (both increasing south to north), respectively (BOM, 2021c, d). Average annual rainfall over the latest 30-year period available (1986–2015) mostly ranges 50–400 mm yr$^{-1}$ (BOM, 2021e), with precipitation falling mostly in winter along the southern fringe of the area whilst the central and northern parts are equally arid year-round. Long-term weather patterns are strongly affected by El Niño/La Niña cycles.

Physiographically, the study area includes, clockwise from the southwest, the Adelaide Hills, North St Vincent–Spencer Gulf, Lower Murray, Lake Frome, Cooper Creek, Bulloo–Bancannia, Warrego, Menindee Lakes, Lachlan, Murrumbidgee, Mid-Murray, Goulburn–Loddon, Wimmera–Mallee, and Millicent Coast physiographic regions (Pain et al., 2011). (Locations of geographical entities named in this paper are given in the Supplement.) The Murray River is Australia's longest river, with its source in the Australian Alps at 1430 m above sea level (a.s.l.) (on the border between NSW and Vic), being approximately 2500 km upstream of its mouth at Goolwa near Victor Harbour (SA) where it enters the Southern Ocean. The combined catchments of the Murray River and that of its northern tributary, the Darling River (the country's third longest river), define the Murray–Darling Basin, a vast (∼ 1 million km$^2$), low-relief Cenozoic depression formed in dynamic topographic response to continental plate motion (Schellart and Spakman, 2015). The Murray–Darling Basin is one of Australia's most productive agricultural areas where water management, critical to support communities and the environment across its mega-catchment, is overseen by the Murray–Darling Basin Authority (MDBA, 2022). Topographic elevation (Geoscience Australia, 2008) in the DCD area ranges from sea level, along the Great Australian Bight region of the Southern Ocean to the southwest, to 1206 m a.s.l., with the highest elevations (>900 m a.s.l.) observed both to the southeast in the Victorian highlands, and to the northwest in the Flinders Ranges; the mean altitude is 139 m a.s.l.

The study area comprises a wide range of geological regions (Blake and Kilgour, 1998; Fig. 2a), the main ones ranging from the Palaeoproterozoic Broken Hill region and Palaeoproterozoic to Mesoproterozoic Mount Painter region, through the Palaeoproterozoic to Devonian Wonominta and Tibooburra regions, the Neoproterozoic to Cambrian Stuart and Adelaide regions, the Neoproterozoic to Ordovician Caloola region, the Cambrian Arrowie, Cambrian to Ordovician Kanmantoo and Cambrian to Devonian Bancannia regions, the Ordovician to Early Carboniferous Lachlan region, the Devonian to Carboniferous Darling region, the Jurassic to Tertiary Eromanga regions, to the Late Mesozoic to Quaternary Murray region. The most dominant surface lithological type is undoubtedly regolith (Eggleton, 2001), the commonly loose (though in places highly indurated), transported or in situ weathering product of primary bedrock, locally modified by aeolian processes. Regolith covers over 85 % of Australia (Wilford, 2012) due to relative tectonic stability, low relief, absence of Quaternary glaciation, low rainfall, and resistant rocks (Pain et al., 2012). Comparatively minor areas of outcrop dot and surround the study area, most notably metamorphic rocks in the central Broken Hill region, sedimentary and felsic to intermediate igneous rocks in the Lachlan region to the east and south, felsic to mafic igneous rocks and metamorphic rocks in the Mount Painter region to the northwest, siliciclastic sediments in the Adelaide and Arrowie regions to the west, and carbonate sediments in the Eromanga region to the north. Very few mafic or ultramafic rocks crop out in the study area, with the main occurrences belonging to the Lachlan region in the southeast.

The soil types (orders) encountered in the study area are most commonly Calcarosol and Vertosol (at 41 % and 33 % of the sample sites, respectively), followed by Sodosol, Kandosol, and Rudosol (8 %, 7 %, and 5 %), and minor Chromosol, Podosol, and Tenosol (∼ 3 % to <1 %), using the Australian Soil Classification scheme (Isbell and the National Committee on Soil and Terrain, 2021; AS-RIS, 2021a) (Fig. 2b). The major river basins that divide the area mostly belong, clockwise from the southwest, to the Murray–Darling, Lake Eyre, and Bulloo–Bancannia drainage divisions, and (in greater detail) to the Lower Murray, Lake Frome, Bulloo–Bancannia, Warrego, Menindee Lakes, Lachlan, Murrumbidgee, Mid-Murray, Goulburn–Loddon, and Wimmera–Mallee water regions (Geoscience Australia, 1997). Land use over the area is overwhelmingly

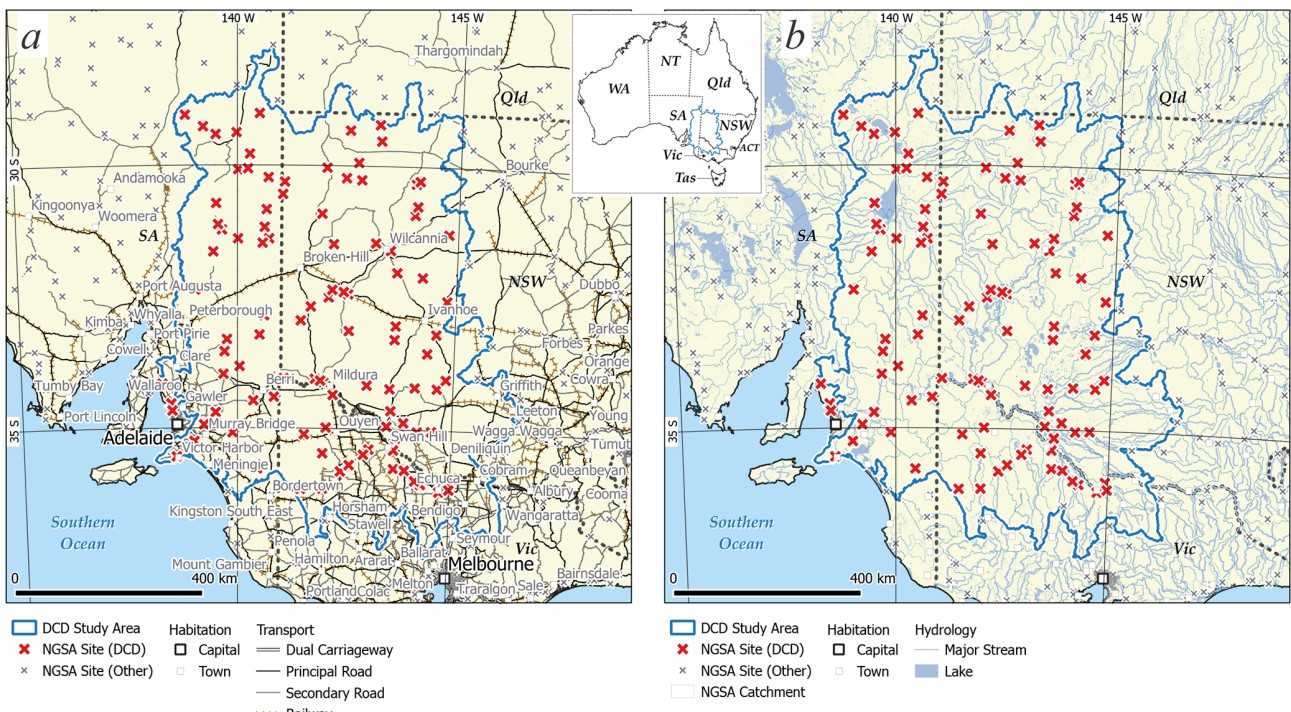

**Figure 1.** The Darling–Curnamona–Delamerian (DCD) Sr isotope study area (dark blue outline, see inset for location: WA – Western Australia, NT – Northern Territory, Qld – Queensland, NSW – New South Wales, ACT – Australian Capital Territory, Vic – Victoria, Tas – Tasmania, and SA – South Australia) and National Geochemical Survey of Australia (NGSA) sample locations (crosses) shown with **(a)** townships (squares), roads (black or grey), and railway (brown); and **(b)** NGSA catchment boundaries (medium blue), major streams, and lakes (pale blue). Map projection: Albers equal area.

grazing native vegetation with minor nature conservation, grazing modified pastures, and dryland cropping (ASRIS, 2021b), the latter two occurring exclusively in the southern part of the study area.

When mapping and interpreting $^{87}$Sr/$^{86}$Sr ratios, it is important to bear in mind the likely processes to may affect them. Pedogenic processes can offset the $^{87}$Sr/$^{86}$Sr of soil compared to that of its parent material as a result of differential dissolution of minerals such as plagioclase and K-feldspar (Blum et al., 1993; Bullen et al., 1997). Although soils in this study were collected on the same parent material, namely fluvial sediment, this effect cannot be excluded. In the same way, it is possible that there is an offset between soils in the sediment source region and the parent rocks they derive from. Dust deposition can also affect the soil Sr budget and shift the $^{87}$Sr/$^{86}$Sr composition of soil compared to that of its parent material (Stewart et al., 1998; Chadwick et al., 2009). Previous work on soils in the study area has shown that in some cases, atmospheric deposition contributes significantly to the soil Sr budget (Green et al., 2004; Mee et al., 2004); in this case, the Sr isotope composition of the soil is not a direct reflection of that of the parent material. However, in other regions of the study area (e.g. Clare Valley), atmospheric deposition plays a much smaller role (Bestland et al., 2003).

## 3 Material and methods

### 3.1 Material

This study utilizes archive "catchment outlet sediment" samples collected during the National Geochemical Survey of Australia (NGSA) which covered $\sim 80\%$ of Australia (de Caritat and Cooper, 2011, 2016). The sampling philosophy of the NGSA was to collect naturally mixed and fine-grained fluvial/alluvial sediments from large catchments, thereby obtaining representative averages of the main soil and rock types contributing sediment through weathering. This allowed the ultralow sampling density ($\sim 1$ sample per catchment which, on average, is 5200 km$^2$) to remain representative of large-scale natural variations (de Caritat and Cooper, 2011). Catchment outlet sediments are similar to floodplain sediments in the sense that they are deposited during receding floodwaters outside the riverbanks but with the added complexity that in Australia, many areas can also experience addition (or loss) of material through aeolian processes. The sampled floodplain geomorphological entities are typically vegetated and biologically active (shrubs, grasses, worms, ants, etc.), thereby making the collected materials true soils, albeit soils developed on transported alluvium parent material.

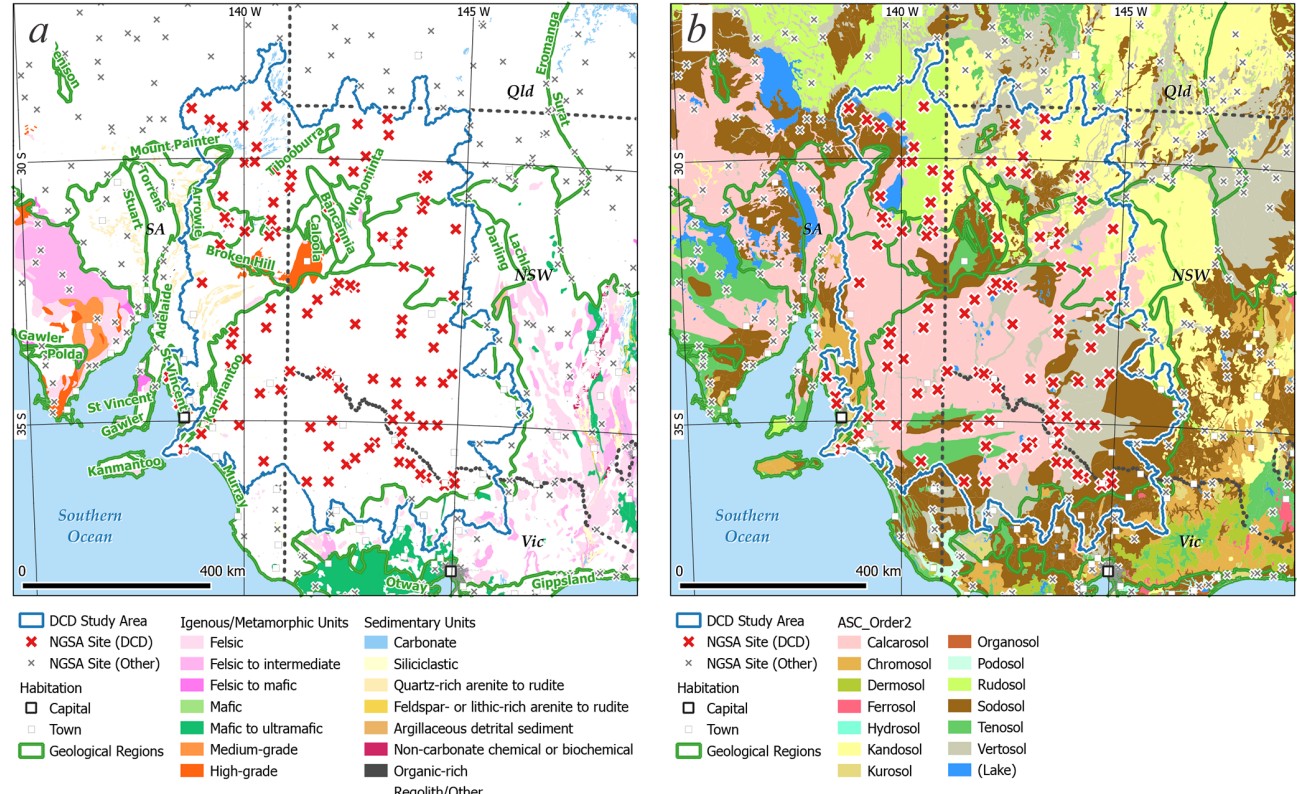

**Figure 2.** The Darling–Curnamona–Delamerian (DCD) Sr isotope study area (dark blue outline), National Geochemical Survey of Australia (NGSA) sample locations (crosses), and geological regions (double green outlines) from Blake and Kilgour (1998) overlain on **(a)** surface geology from Raymond et al. (2012); and **(b)** Australian Soil Classification soil orders from Isbell and the National Committee on Soil and Terrain (2021). Map projection: Albers equal area.

The sampling medium and density were both strategically chosen in the NGSA project to prioritize coverage over resolution. This was justified by the fact that the NGSA was Australia's first and, to date, only fully integrated, internally consistent geochemical survey with a truly national scope. In terms of the DCD, it is clear that these choices have implications on the granularity of the patterns revealed by the Sr isoscape; as the collection of Sr isotope data in Australia using NGSA samples grows in the future, it is hoped the value of coverage will prevail over a relative low resolution of detailed features.

The NGSA collected samples at two depths, a "top outlet sediment" (TOS) from a shallow (0.1 m) soil pit approximately 0.8 m × 0.8 m in area, and a "bottom outlet sediment" (BOS) from a minimum of three auger holes generally drilled within ∼ 10 m of the TOS pit. The auger holes were drilled to refusal or to maximum depth of ∼ 1 m, and the BOS sample was collected, on average, from a top depth of 0.6 m to bottom depth of 0.8 m from all augered holes. A field manual (Lech et al., 2007) was compiled to record all sample collection method details, including the site selection process. Sampling for the NGSA took place between July 2007 and November 2009, and the field data were recorded in Cooper et al. (2010).

In the laboratory, the samples were air dried at 40 °C for a minimum of 48 h (or to constant mass), homogenized and reduced by riffle splitting, sieved to two different grain-size fractions (a coarse fraction of <2 mm and a fine fraction of <75 μm), and further prepared (see de Caritat et al., 2009) for the comprehensive geochemical analysis programme of the NGSA (see de Caritat et al., 2010). For this Sr isotope analysis, an aliquot of minimum ∼ 1 g of sample was milled to a fine powder using a carbon steel ring mill or, for a few samples only, an agate micromill. The main sample type selected was BOS <2 mm to be as representative as possible of the geogenic background unaffected by modern land use practices and inputs (e.g. fertilizers).

Overall, 112 NGSA BOS <2 mm samples were analysed for Sr isotopes as detailed in the Methods subsection below. Those samples originate from within 102 NGSA catchments, which together cover 529 000 km$^2$ of inland southeastern Australia (see Fig. 1).

## 3.2 Methods

Samples were prepared and analysed for Sr isotopes ($^{87}$Sr/$^{86}$Sr) at the Wollongong Isotope Geochronology Laboratory (WIGL). Approximately 50 mg of sample was weighed and digested in a 2:1 mixture of hydrofluoric and nitric acids. All reagents used were Seastar Baseline® grade, with Sr concentrations typically <10 parts per trillion. Following digestion, samples were re-dissolved in aqua regia (twice if needed) in order to eliminate any fluorides, followed by nitric acid twice. Finally, samples were re-dissolved in 2 M nitric acid prior to ion exchange chromatography. Strontium was isolated from the sample matrix using automated, low-pressure chromatographic system Elemental Scientific prepFAST-MC™ and a 1 mL Sr–Ca column (Eichrom™) (Romaniello et al., 2015). The Sr elutions were re-dissolved in 0.3 M HNO$_3$. Strontium isotope analysis was performed on a Thermo Scientific Neptune Plus multicollector-inductively coupled plasma mass spectrometer (MC-ICP-MS) at WIGL. The sample introduction system consists of an ESI Apex-ST PFA MicroFlow nebulizer with an uptake rate of $\sim 0.1$ mL min$^{-1}$, an SSI Quartz dual cyclonic spray chamber, jet sample, and X-skimmer cones. Measurements were performed in low-resolution mode. The instrument was tuned at the start of each session with a 20 parts per billion Sr solution, and sensitivity for $^{88}$Sr was typically around 4 V. Masses 88, 87, 86, 85, 84, and 83 were collected on Faraday cups. Instrumental mass bias was internally corrected using measured $^{88}$Sr/$^{86}$Sr. Masses 85 and 83 were used to correct for the isobaric interference of $^{87}$Rb and $^{86}$Kr, respectively.

Maps were prepared using the open software QGis (version 3.16.14–Hannover) and applying an Albers equal area projection. Symbology for displaying $^{87}$Sr/$^{86}$Sr data was either point data classified in eight equal quantile classes (12.5 % of the data each; green = low to high = red) at the sampling site, or attributing this same value and colour to the whole catchment from which the outlet sediment comes, reflecting the sampling medium, catchment outlet sediment, being a representative sample of the average materials in the catchment (see Sect. 3.1).

## 3.3 Quality assessment

National Institute of Standards and Technology (NIST) strontium carbonate isotope Standard Reference Material SRM987 was used as a secondary standard and measured after every five samples to assess accuracy during analysis. Accuracy of the whole procedure was assessed by processing United States Geological Survey (USGS) reference material Basalt from the Columbia River standard BCR-2 (Plumlee, 1998). The mean $\pm 2$se $^{87}$Sr/$^{86}$Sr for BCR-2 in this study is 0.704961 ± 35 ($n = 13$), within error of the value in Jweda et al. (2016) (0.704500 ± 11). Total procedure blanks ranged between 0.025 and 0.245 ng Sr ($n = 12$).

Ten field duplicate sample pairs (collected at a median distance of $\sim 80$ m from one another on the same landscape unit, see Lech et al., 2007) were analysed for $^{87}$Sr/$^{86}$Sr in the BOS <2 mm sample, and returned a median relative standard deviation of 0.10 %. The relative standard deviation from field duplicates includes natural variability (mineralogical/chemical heterogeneity of the alluvial deposit), as well as sample collection, preparation, and analysis uncertainties.

Overall, we feel that the quality of the $^{87}$Sr/$^{86}$Sr data presented herein is adequate for the purpose of regional mapping, and that reporting $^{87}$Sr/$^{86}$Sr data to the third decimal place with an indicative fourth decimal place is appropriate for this work. This relatively low precision is attributed to heterogeneity of the alluvial deposits, since precision relating to sample preparation and analysis for Sr isotopes is at the fifth decimal place (see results for BCR-2 above).

## 4 Results and discussion

The DCD region $^{87}$Sr/$^{86}$Sr results are summarized in Table 1 and the full dataset ($n = 112$) is available from the https://portal.ga.gov.au/restore/cd686f2d-c87b-41b8-8c4b-ca8af531ae7e (last access: 22 August 2022) and from de Caritat et al. (2022). The observed $^{87}$Sr/$^{86}$Sr values range from a minimum of 0.7089 to a maximum of 0.7511 (range 0.0422). The median $^{87}$Sr/$^{86}$Sr ($\pm$ median absolute deviation) result is 0.7199 ($\pm 0.0071$) and the mean ($\pm$ standard deviation) is 0.7220 ($\pm 0.0106$). Kurtosis and skewness are 0.28576 and 1.0285, respectively. The whole-dataset distribution characteristics indicate a bimodal to polymodal platykurtic (i.e. displaying few outliers) distribution with a heavy positive (high) tail (Fig. 3). Indeed, the Tukey boxplot (Tukey, 1977) shows only one structural outlier at the maximum value. The modes of the two main subpopulations are $\sim 0.712$ and $\sim 0.721$, with several smaller subpopulations having more radiogenic values. We note that other large-scale $^{87}$Sr/$^{86}$Sr datasets have a similar range to the DCD region, are also at least bimodal, and also have heavily right-hand (positively) skewed distributions (e.g. Bataille et al., 2020). Statistics for individual geological regions are also shown in Table 1.

The observed $^{87}$Sr/$^{86}$Sr values range from close to modern seawater $^{87}$Sr/$^{86}$Sr (0.709), also reflecting modern marine carbonate rock and mudstone, to values characteristic of older, Rb-rich felsic igneous rocks such as granite and rhyolite; the median and mean values are typical of monzogranite to quartzite main rock types (Bataille and Bowen, 2012, Fig. 5). In the following subsections, we discuss potential sources for the more elevated $^{87}$Sr/$^{86}$Sr values (top quartile) in the area, followed by an analysis of potential depleted sources (bottom quartile), the variation of $^{87}$Sr/$^{86}$Sr along the Murray River corridor, the variation of $^{87}$Sr/$^{86}$Sr in different geological regions within the DCD area, and fi-

**Table 1.** Summary statistics (count, minimum, 25th percentile, median, median absolute deviation, mean, standard deviation, 75th percentile, maximum, range, variance, inter-quartile range, kurtosis, and skewness) of the $^{87}$Sr/$^{86}$Sr data from the Darling–Curnamona–Delamerian (DCD) Sr isotope study area, southeastern Australia. NA: not available.

| Georegion | $n$ | Min | 25 % | Med | MAD | Mean | SD | 75 % | Max | Range | Var | IQR | Kurt | Skew |
|---|---|---|---|---|---|---|---|---|---|---|---|---|---|---|
| All | 112 | 0.7089 | 0.7129 | 0.7199 | 0.0071 | 0.7220 | 0.0106 | 0.7281 | 0.7511 | 0.0422 | 0.0001 | 0.0152 | 0.2856 | 1.0285 |
| Adelaide | 7 | 0.7162 | 0.7200 | 0.7244 | 0.0062 | 0.7264 | 0.0082 | 0.7325 | 0.7402 | 0.0240 | 0.0001 | 0.0124 | −0.2214 | 0.6163 |
| Arrowie | 2 | 0.7201 | 0.7201 | 0.7248 | 0.0043 | 0.7248 | 0.0066 | 0.7294 | 0.7294 | 0.0093 | $4.35 \times 10^{-5}$ | 0.0093 | NA | NA |
| Bancannia | 1 | 0.7180 | 0.7180 | 0.7180 | NA | 0.7180 | NA | 0.7180 | 0.7180 | NA | NA | NA | NA | NA |
| Broken Hill | 4 | 0.7239 | 0.7267 | 0.7365 | 0.0070 | 0.7369 | 0.0109 | 0.7474 | 0.7506 | 0.0266 | 0.0001 | 0.0207 | 1.1939 | 0.1979 |
| Darling | 8 | 0.7102 | 0.7113 | 0.7152 | 0.0035 | 0.7152 | 0.0040 | 0.7189 | 0.7212 | 0.0110 | $1.57 \times 10^{-5}$ | 0.0076 | −1.0963 | 0.2907 |
| Eromanga | 26 | 0.7089 | 0.7115 | 0.7142 | 0.0034 | 0.7178 | 0.0093 | 0.7220 | 0.7445 | 0.0356 | 0.0001 | 0.0105 | 2.1347 | 1.6451 |
| Kanmantoo | 2 | 0.7495 | 0.7495 | 0.7497 | 0.0002 | 0.7497 | 0.0003 | 0.7499 | 0.7499 | 0.0004 | $9.33 \times 10^{-8}$ | 0.0004 | NA | NA |
| Lachlan | 5 | 0.7246 | 0.7254 | 0.7284 | 0.0023 | 0.7320 | 0.0109 | 0.7403 | 0.7511 | 0.0265 | 0.0001 | 0.0149 | 4.4042 | 2.0653 |
| Murray | 54 | 0.7095 | 0.7129 | 0.7195 | 0.0065 | 0.7214 | 0.0094 | 0.7261 | 0.7419 | 0.0324 | 0.0001 | 0.0133 | −0.2846 | 0.8889 |
| St Vincent | 1 | 0.7312 | 0.7312 | 0.7312 | NA | 0.7312 | NA | 0.7312 | 0.7312 | NA | NA | NA | NA | NA |

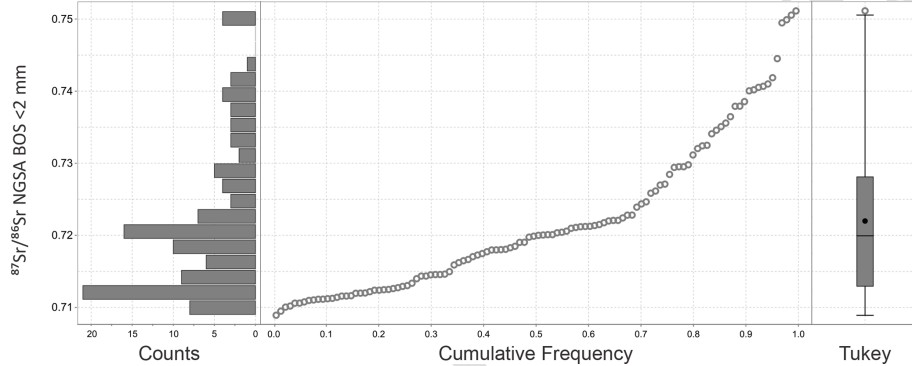

**Figure 3.** Univariate distribution of the $^{87}$Sr/$^{86}$Sr data from the Darling–Curnamona–Delamerian (DCD) Sr isotope study area: (left) histogram (20 bins 0.002 wide); (middle) cumulative frequency plot; and (right) Tukey boxplot (mean = black dot).

nally compare the $^{87}$Sr/$^{86}$Sr data to existing geochemical and mineralogical data for the same NGSA sites.

## 4.1 Sr isoscape mapping

From the $^{87}$Sr/$^{86}$Sr value reported at any given sampling location (Fig. 4a), we can infer a representative catchment $^{87}$Sr/$^{86}$Sr composition for the drainage area directly upstream of it (Fig. 4b) because sediments sampled at this location (outlet of catchment) represent an average of sediments eroded upstream and transported to the sampling location (Ottesen et al., 1989; Bølviken et al., 2004). This approach allows cost-effective geochemical or isotopic mapping of a large area despite a low sampling resolution. However, in addition to the limitations relating to weathering and aeolian inputs mentioned above, one needs to bear in mind that this method assumes that all areas of the landscape draining into the sampling location contribute equally to the sediment budget, which may not be accurate. These limitations need to be taken into consideration before using the reported $^{87}$Sr/$^{86}$Sr as proxies for the Sr isotope composition of the basement in the mapped region. If using this dataset to study past or present animal mobility, another limitation is that we mea-

sured the isotopic composition of bulk Sr in sediments, not the bioavailable Sr. Nevertheless, the dataset presented here can still be used as a guide or indeed an input to machine learning models, for geological, archaeological, palaeontological, and ecological studies.

## 4.2 Identification of radiogenic $^{87}$Sr/$^{86}$Sr source regions

Spatial patterns observed on the $^{87}$Sr/$^{86}$Sr map (Fig. 4) indicate inputs and transport of elevated $^{87}$Sr/$^{86}$Sr signatures down several waterways. The most prominent such "train" of dispersion is along the Murray River that forms the state border between NSW and Vic in the southeastern part of the study area. Here, several $^{87}$Sr/$^{86}$Sr values range from 0.7274 to 0.7511 (i.e. in the top quartile of the dataset), including the maximum value from this survey. On closer inspection, it is evident that the elevated $^{87}$Sr/$^{86}$Sr signatures come down not only the Murray River (three samples), but also from tributaries that join the Murray River and carry sediments down catchments from upland areas both south and north of the border river. Examples of such streams are (from east to west) the Goulburn, Bendigo, and Loddon rivers/creeks lo-

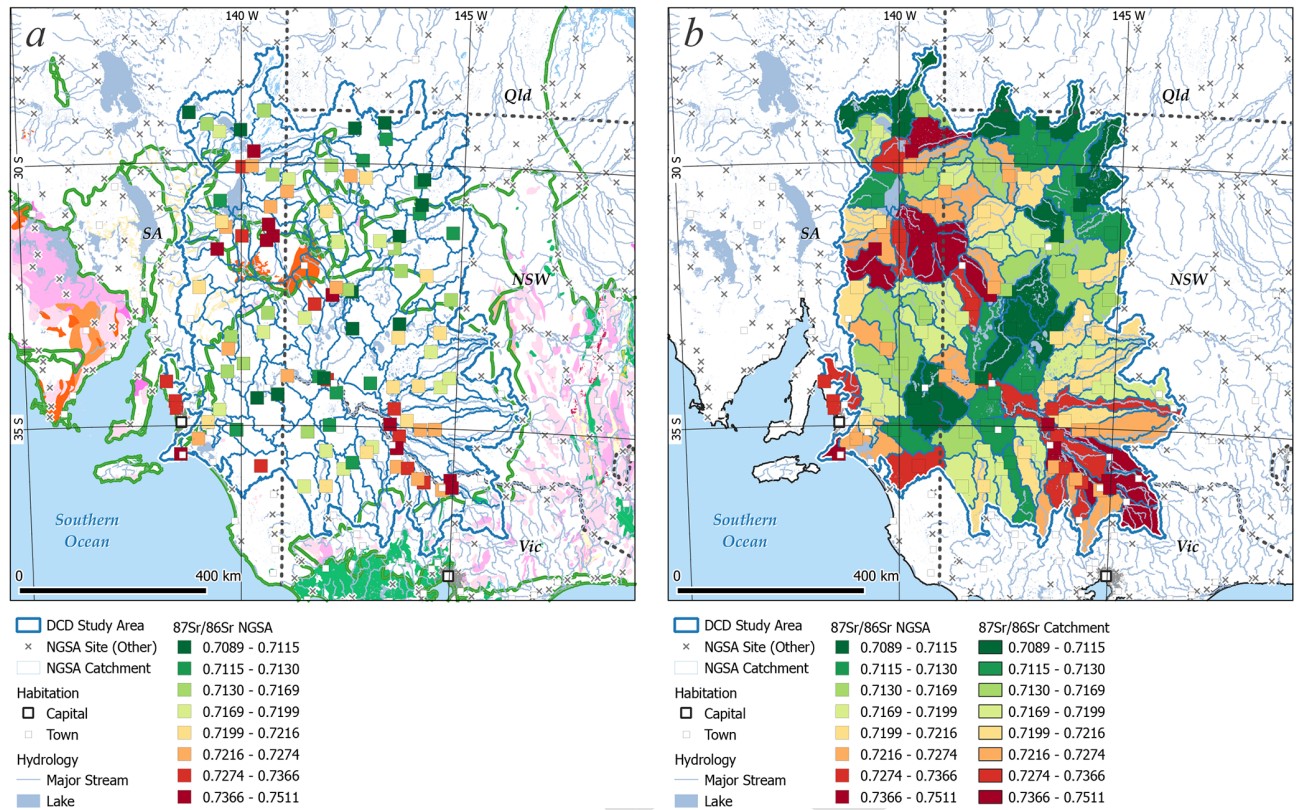

**Figure 4.** Strontium isoscape for the Darling–Curnamona–Delamerian (DCD) Sr isotope study area with data points classed by quantiles and overlain **(a)** on geological regions (double green outlines) from Blake and Kilgour (1998) and surface geology from Raymond et al. (2012) (see Fig. 2a for geology legend); and **(b)** on catchments coloured by same colour ramp. Data are classified in eight equal quantile classes. Map projection: Albers equal area.

cated south of the Murray River, and the Wakool, Edward, and Murrumbidgee rivers/creeks to the north of it (see Supplement). Both the Murray and tributary rivers drain abundant felsic igneous rocks hosted in the Ordovician turbidites of the Lachlan Fold Belt to the east and in the Victorian highlands to the south and southeast (both part of the Lachlan geological region), which are likely candidates for sourcing the elevated $^{87}$Sr/$^{86}$Sr minerals transported down catchment. Present-day $^{87}$Sr/$^{86}$Sr ratios for those sediments and intrusions have indeed been reported around and over 0.72 (e.g. Keay et al., 1997; Chappell et al., 2000; Hergt et al., 2007). Minor mafic lithologies, e.g. at Dookie east of Shepparton (Broken Creek catchment) with lower $^{87}$Sr/$^{86}$Sr reported values of 0.704 to 0.707 (Foster et al., 2009), are not of sufficient extent to significantly dampen the elevated $^{87}$Sr/$^{86}$Sr signature at the catchment outlet.

A lone similarly elevated (top quartile) sample is located in the south central part of the study area and comes from a dune-dominated landscape devoid of principal drainage (Ngarkat Conservation Park in SA). Only a couple of very small areas of Cambrian–Ordovician felsic igneous rocks crop out in the west of this catchment, possibly suggesting that the elevated $^{87}$Sr/$^{86}$Sr signal is inherited by aeolian in-

put from source region(s) further afield. Indeed, the strongest average winds over the period 2004–2008 at this site, which tend to occur in winter, are from the northwest (BOM, 2021f). This upwind direction has abundant Adelaidean-age (meta)sediments, felsic, and intermediate igneous rocks, and medium- and high-grade metamorphic rocks ($^{87}$Sr/$^{86}$Sr commonly >0.73), e.g. on the Yorke and Eyre peninsulas, and in the Gawler geological region, which are possible sources of elevated $^{87}$Sr/$^{86}$Sr mineral grains potentially transported as windblown dust (Fig. 4; Foden et al., 2001; Bestland and Forbes, 2009).

Along the Southern Ocean coast both north and south of Adelaide, four top-quartile samples occur along the Inman, Gawler, Light, and Wakefield rivers, which are in more urbanized/industrialized environments compared to the rest of the study area. These samples are likely also influenced by aeolian transport and deposition of radiogenic material from the Yorke, Eyre, and Gawler source areas mentioned above.

South and north of the Broken Hill geological region near the centre of the study area, similarly elevated $^{87}$Sr/$^{86}$Sr values ($\geq 0.7274$) are observed in several catchments: Pine and Stephens creeks to the south, and Siccus River, and Digby, Jack, and Childs dam channels (the latter three are from low-

relief and featureless landscapes with poorly defined natural drainages), and Eurinilla Creek to the north. All of these catchments drain felsic igneous and/or high-grade metamorphic lithologies in the Palaeoproterozoic Broken Hill domain that would likely be the sources of elevated $^{87}$Sr/$^{86}$Sr signatures (Pidgeon, 1967; Page and Laing, 1992) in those sediments. The radiogenic signature of the Broken Hill basement rocks can also be recognized in the regional groundwater (de Caritat et al., 2005).

Two samples north of Lake Frome also have top-quartile $^{87}$Sr/$^{86}$Sr signatures. One of them is located at the bottom of the catchment draining Mount Painter in the northeastern Flinders Ranges, an area of felsic igneous and medium-grade metamorphic rocks, which would contain prime candidate lithologies to produce sediment with a radiogenic Sr isotope signature (Elburg et al., 2003; McLaren et al., 2006). The second sample, located near Bakers Bore at the edge of the Strzelecki Desert, in a field of southwest–northeast oriented linear dunes, likely represents the $^{87}$Sr/$^{86}$Sr signature of wind-transported material from the same Mount Painter source region and/or the outcropping Proterozoic rocks of the Broken Hill domain discussed above, broadly consistent with dominant wind directions (BOM, 2021f).

## 4.3 Identification of unradiogenic $^{87}$Sr/$^{86}$Sr source regions

Whereas attention above has been paid to identifying the potential source rocks or regions for the more radiogenic $^{87}$Sr/$^{86}$Sr catchment sediment signatures observed in the DCD study, it is also instructive to look at the catchments with lower $^{87}$Sr/$^{86}$Sr. Catchments with $^{87}$Sr/$^{86}$Sr values $< \sim$ 0.7130 (i.e. in the bottom quartile of the dataset) occur in the central Murray Basin away from the main Murray River. To the south of the river, several low-relief (essentially internally draining) catchments with virtually no bedrock outcrop have $^{87}$Sr/$^{86}$Sr values (from west to east) at 0.7129 and 0.7126 (field duplicates; Karoonda catchment), 0.7112 (Buniyp Plain catchment), 0.7095 (Berri catchment), and 0.7120 (Callaghan Plain catchment). The Callaghan Plain catchment is abutted to the south by the Ouyen catchment ($^{87}$Sr/$^{86}$Sr = 0.7116) and the Woomelang catchment to its south (0.7122); both of which are also low relief, locally featuring dunes and/or man-made drainage channels rather than natural stream lines. All of these $^{87}$Sr/$^{86}$Sr values are at the lower end of the typical Murray River sediments, as discussed in the next subsection.

To the north of the Murray River, the Great Darling Anabranch and Darling River have signatures of 0.7120, and 0.7116 and 0.7114 (field duplicates), respectively. The Arumpo catchment has a value of 0.7127, whilst to its north, the Karoola, Garnpung, and Talyawalka catchments have $^{87}$Sr/$^{86}$Sr values of 0.7114, 0.7110, and 0.7112, respectively. All of these catchments lack mapped rock outcrop, and consist therefore almost entirely of Darling Basin sediment, with an unknown potential contribution from aeolian material in places. This is entirely consistent with the assessment of Gingele and De Deckker (2005) that typical source rocks in the New England Fold Belt (where Darling River tributaries mainly originate from) are Tertiary basalts with $^{87}$Sr/$^{86}$Sr ratios of 0.703–0.705 and Permian granites with ratios around 0.7127.

In the northeast of the DCD study area, there are several catchments also with low $^{87}$Sr/$^{86}$Sr signatures: 0.7102 (Paroo Overflow), 0.7124 (Tambua), 0.7116 (Paroo Overflow 2), 0.7108 (Cuttaburra), 0.7125 (Paroo), and 0.7106 (Cuttaburra Channels) toward the north–northeast from Wilcannia. In the central north, four further catchments have lower quartile $^{87}$Sr/$^{86}$Sr values: Caryapundy (0.7111), Bootra (0.7120), Feeder (0.7125), and Berawinnia (0.7106) catchments.

Finally, in the northwest of the study area where the Eromanga geological region encroaches onto the DCD area, a sand dune-dominated catchment on the edge of the Strzelecki Desert east of Lake Blanche has low $^{87}$Sr/$^{86}$Sr (0.7100), as does the Cooryanna catchment (0.7089) further west.

As the Murray–Darling Basin is a region of locally intense agricultural activity, the possibility of fertilizers affecting the present $^{87}$Sr/$^{86}$Sr results needs to be considered. Fertilizers used in a subcatchment of the Namoi River basin located in the eastern Murray–Darling Basin (to the east of the DCD study area), have been documented to have $^{87}$Sr/$^{86}$Sr values in the range 0.7081 to 0.7090, and application rates up to 30 to 60 kg ha$^{-1}$ yr$^{-1}$ (Martin and McCulloch, 1999). These values fall within the lowest 12.5 % of our dataset (darkest green colour in Fig. 4). Although we cannot rule out some localized influence of fertilizers on the documented $^{87}$Sr/$^{86}$Sr patterns, it is worth remembering that (1) the dominant land uses in the DCD region are overwhelmingly grazing native vegetation (with minor nature conservation, grazing modified pastures, and dryland cropping), as noted in Sect. 2 above; and (2) our choice of sampling subsoil rather than topsoil material to reduce the impact of anthropogenic actions and practices together should minimize that potential influence.

To close this subsection, we note that the unradiogenic $^{87}$Sr/$^{86}$Sr regolith values discussed above are consistent with regional, background $^{87}$Sr/$^{86}$Sr groundwater values reported from shallow aquifers in the Flinders Ranges, Lake Frome, and Eromanga Basin regions (Ullman and Collerson, 1994; de Caritat et al., 2005).

## 4.4 Murray River profile

A profile of the catchment sediments' $^{87}$Sr/$^{86}$Sr value along the Murray River (and tributaries) is shown in Fig. 5 for sampling sites within a 5 km buffer and a 10 km buffer from the river. The profile indicates a marked decrease in $^{87}$Sr/$^{86}$Sr from the highlands (0.74–0.75) to near the mouth of the river ($\sim$ 0.71), spanning nearly the whole data range for the DCD study area. The 5 km and the 10 km buffered sampling points give linear regressions with similar correlation coefficients

($R = -0.67$ and $-0.66$, respectively), and identical slopes (both $-3 \times 10^{-5}$). Both regressions also suggest a primary $^{87}Sr/^{86}Sr$ of source material at the headwaters of the Murray River (intercept) of 0.749–0.751, consistent with the isotopic signature of felsic igneous and sedimentary rocks in the surrounding Lachlan geological region (e.g. Keay et al., 1997; Chappell et al., 2000; Hergt et al., 2007).

Along this profile, a few tributaries have clearly anomalous $^{87}Sr/^{86}Sr$ signatures; for instance, the Goulburn River and the Campaspe River in the highlands are much higher ($\sim 0.751$) and lower ($\sim 0.726$), respectively, than the Murray River sediment signature in this part of the basin ($\sim 0.740$). Further downstream, the numerous channels and low-relief landscape from the Callaghan Plain contribute low $^{87}Sr/^{86}Sr$ material to the main Murray valley from the south. Next, the Darling River confluence also contributes less radiogenic sediments ($\sim 0.711$) from the north.

These findings are consistent with the study of the fluvial suspended particulate matter by Douglas et al. (1995), which found more radiogenic values in clay and coarser ($>0.2\,\mu m$) fraction material from the Murray River ($\sim 0.722$ to 0.760) than from the Darling River ($\sim 0.709$–0.711). Gingele and De Deckker (2005) studied the Sr isotopic composition of fluvial clay-sized sediments ($<2\,\mu m$) of the Murray and Darling basins, and also found the former to be more radiogenic and variable (0.726–0.775) than the latter (0.708–0.717). Aeolian dust collected in the Murray and Darling basins (De Deckker et al., 2014) similarly shows a radiogenic divide between these source regions, with Murray Basin dust spanning a wide range of more radiogenic $^{87}Sr/^{86}Sr$ (0.716–0.776) compared to the Darling Basin dust (0.708–0.714).

Further downstream, closest to the mouth of the river, sediments are being transported down from the west with a more radiogenic signature ($\sim 0.720$) relative to the Murray River sample(s) further upstream; this signature is consistent with rocks from the Kanmantoo geological region (Gray, 1990; Bestland and Forbes, 2009; Haines et al., 2009) shedding sediment down Reedy Creek.

## 4.5   Isotopic signature of the geological regions

In consideration of the above discussion on more radiogenic and less radiogenic source regions for the minerals found in the NGSA sediments of the DCD, Fig. 6 shows a clear split between these regions in the $^{87}Sr/^{86}Sr$ vs. 1000/Sr variable space. By and large, the sediments provenanced from, or deposited in, the Darling Basin and Eromanga Basin (only the southern part of which is considered in this study) appear to have relatively low $^{87}Sr/^{86}Sr$ ratios (mostly $<0.7250$), in agreement with previous studies cited above. At the same time, these sediments have a widely variable Sr content (1000/Sr ranging from 1.2 to 33.6 $g\,\mu g^{-1}$). In con-

trast, NGSA sediments collected in catchments draining[1] the other main geological regions represented in the DCD area (Adelaide, Arrowie, Broken Hill, Kanmantoo, Lachlan, and Murray geological regions; open symbols in Fig. 6) can have more radiogenic $^{87}Sr/^{86}Sr$ signatures (up to 0.750). These sediments also have a more limited 1000/Sr (1000/Sr ranging mostly from 0.4 to $\sim 21\,g\,\mu g^{-1}$). Generalized trends for these two populations have been suggested by the qualitative arrows in the scatterplot.

Three samples from the Murray geological region, however, have much lower Sr contents (1000/Sr between 35 and 75 $g\,\mu g^{-1}$) than the others (Fig. 6); closer examination of these unusual cases indicates that they are from sandy deserts near the Vic–SA border in the southwestern DCD area (Boinka at the southern edge of Sunset National Park, Ngarkat in the Ngarkat Conservation Park, and Wirrengren in the Big Desert). Here, aeolian input of quartz grains may be or may have been likely active, diluting the geochemical and mineralogical concentrations of Sr-bearing components in the samples. Indeed, these three sites contain over 90 wt % $SiO_2$, which is over the 75th percentile nationally for the NGSA (de Caritat and Cooper, 2011), and among the highest silica content of all the DCD samples. Since the aeolian quartz input is likely exogenous (from a different source to the provenance of the silt and clay material in the sample), we can reasonably speculate that the fluvial component of these materials would be significantly richer in Sr, and therefore have lower 1000/Sr values.

Three catchment samples draining mainly the Eromanga geological region, conversely, fall within the general Adelaide–Broken Hill, etc. trend (open symbols and arrow in Fig. 6). These come from the Broken Hill, Lake Callabonna, and Strzelecki Desert catchments in the central-northern part of the study area. The samples are located either downstream from the Broken Hill highlands or downwind from the Mount Painter and Adelaide regions. They are thus likely influenced by these more radiogenic source regions even though physically located in the Eromanga geological region, which typically has a lower $^{87}Sr/^{86}Sr$ sediment signature.

To refine the differentiation of $^{87}Sr/^{86}Sr$ signature for the various geological regions, boxplots of the data subgrouped by these regions are presented in Fig. 7. In Fig. 7a, Tukey boxplots are shown first for the whole dataset ($n = 112$), then in succession by the drained geological region in order of increasing median (or, in two cases, unique) value: Darling ($n = 8$), Wonominta (2), Eromanga (26), Bancannia (1), Murray (54), Arrowie (2), Adelaide (7), St Vincent (1), Lachlan (5), Broken Hill (4), and Kanmantoo (2) geological regions. A histogram indicating the contribution of each geological region to the general population distribution is shown

---

[1]Samples are attributed a geological region here (Figs. 6 and 7) not based on which region they are necessarily located in (sample site), but based on which region is most represented in their catchment.

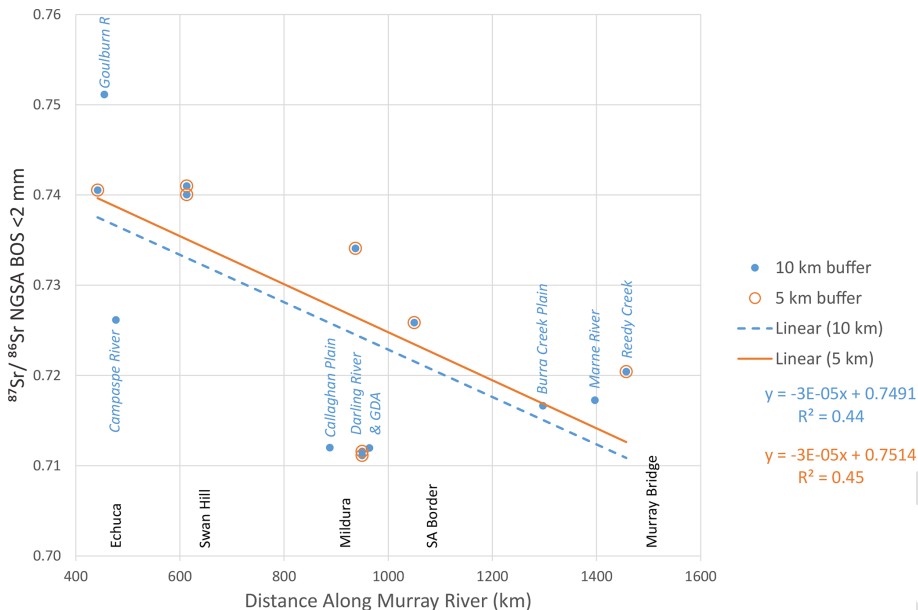

**Figure 5.** $^{87}$Sr/$^{86}$Sr profile of NGSA catchment sediment at sampling sites located within 5 and 10 km buffers of the Murray River (open and closed circles, respectively) vs. downstream distance along the Murray River (east to west). Sites not directly sampling a Murray River catchment are labelled (in italics) with the name of the tributary (R = River; GDA = Great Darling Anabranch) or geomorphic entity. Least squares regression lines for the 5 and 10 km buffers are shown as solid and dashed lines, respectively, and equations are given. Township and South Australia (SA) border positions are shown on the *x* axis.

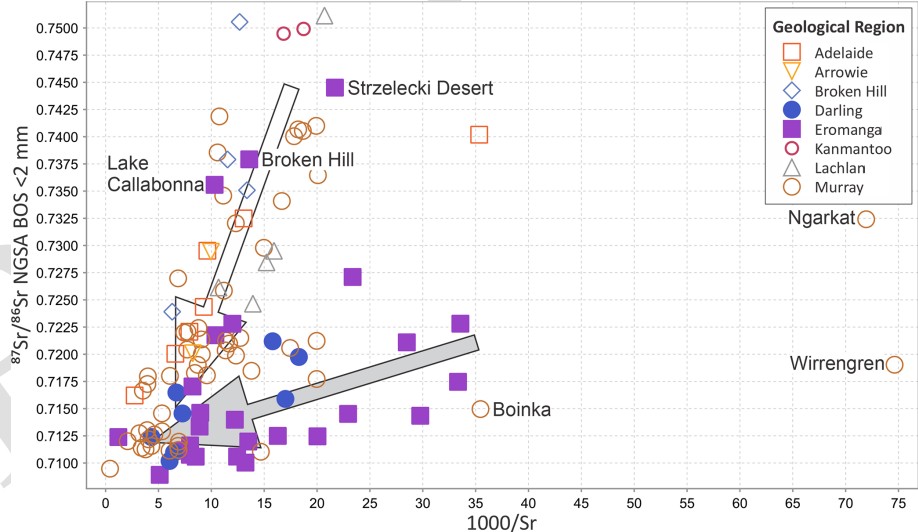

**Figure 6.** Scatterplot of $^{87}$Sr/$^{86}$Sr vs. 1000/Sr in the NGSA BOS <2 mm samples from the Darling–Curnamona–Delamerian (DCD) Sr isotope study area, grouped by main geological region each catchment is draining.

in Fig. 7b. With the few exceptions explained above, both diagrams support a general model with two main sources of fluvial and aeolian sediment in the DCD area, a more radiogenic source ($^{87}$Sr/$^{86}$Sr between 0.720 and 0.751) consisting generally of the older (Palaeoproterozoic to Early Carboniferous) geological regions (Kanmantoo, Broken Hill, Lachlan, Adelaide, and Arrowie) versus a less radiogenic source

($^{87}$Sr/$^{86}$Sr between 0.709 and 0.722) consisting of generally younger (Devonian to Tertiary) geological regions (Eromanga and Darling). The most expansive and youngest (Late Mesozoic–Quaternary) Murray geological region represents a transition between these two endmembers, as, depending on sampling location, it is the depositional site for material

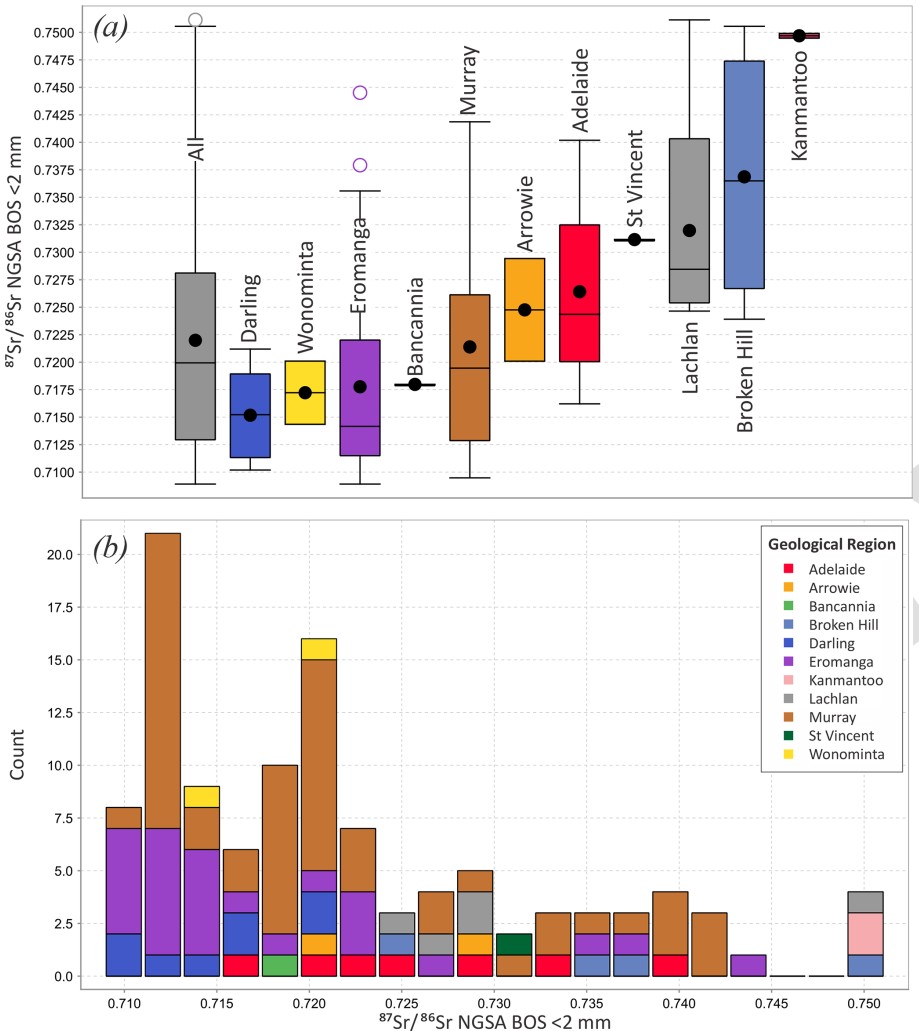

**Figure 7.** Univariate distribution representations of the $^{87}$Sr/$^{86}$Sr data from the Darling–Curnamona–Delamerian (DCD) Sr isotope study area by geological region: **(a)** Tukey boxplots of, left to right, the whole dataset first (grey; $n = 112$), then sorted by increasing median (Darling, $n = 8$; Wonominta, $n = 2$; Eromanga, $n = 26$; Bancannia, $n = 1$; Murray, $n = 54$; Arrowie, $n = 2$; Adelaide, $n = 7$; St Vincent, $n = 1$; Lachlan, $n = 5$; Broken Hill, $n = 4$; and Kanmantoo, $n = 2$, geological regions); and **(b)** histogram by the above subgroups.

mainly contributed by either the radiogenic or the unradiogenic endmember source.

The smaller Palaeoproterozoic–Devonian Wonominta region, drained by two of the catchments sampled here, yields
[5] $^{87}$Sr/$^{86}$Sr values (0.7144 and 0.7201) uncharacteristic of the contemporaneous radiogenic endmember identified above, but intermediate between it and the Mt Wright metavolcanics of alkali basalt affinity (0.704–0.706; Plimer, 1985; Zhou and Whitford, 1994) found in the Wonominta block. In the ab-
[10] sence of published $^{87}$Sr/$^{86}$Sr data for other lithologies, particularly felsic volcanic and (meta)sedimentary lithologies, from the Wonominta block, it is impossible to determine if the two catchment sediment values reported here are typical of a weighted average Wonominta block Sr isotope signature
[15] or, instead, represent a mixture of material originating from it and from either of the two endmembers discussed above.

## 4.6 Comparison to geochemistry and mineralogy

The $^{87}$Sr/$^{86}$Sr values in the DCD area discussed here are loosely correlated with the concentrations of major elements/constituents potassium (Spearman correlation coeffi-
[20] cient $r_S = 0.35$), calcium ($-0.33$), and loss on ignition (a proxy for organic matter; $-0.32$) in the same BOS coarse NGSA sediment samples (de Caritat and Cooper, 2011). (Note that for $n = 112$, $|r_S|$ values $>0.24$ have $p = 0.001$.) A number of trace elements similarly have noteworthy cor-
[25] relations with the $^{87}$Sr/$^{86}$Sr values in the DCD area, most notably the total concentrations of thorium (0.48), tungsten (0.44), lanthanum (0.42), and light rare earth elements praseodymium, cerium, and neodymium (0.38 to 0.35), rubidium (0.35), lead (0.35), and Sr itself ($-0.52$) (de Car-
[30] itat and Cooper, 2011). In terms of bulk mineralogy, the

$^{87}$Sr/$^{86}$Sr values of the DCD samples correlate best with the abundances of plagioclase (0.44) and illite–muscovite (0.43) in the TOS bulk NGSA sediments (de Caritat and Troitzsch, 2021). Therefore, weak spatial distribution similarities exist between the $^{87}$Sr/$^{86}$Sr maps and the geochemical/ mineralogical maps for the above parameters.

## 5   Data availability

The new spatial Sr isotope dataset for the DCD region is publicly available through the https://portal.ga. gov.au/restore/cd686f2d-c87b-41b8-8c4b-ca8af531ae7e (last access: 22 August 2022) and from https://doi.org/10.26186/146397 (de Caritat et al., 2022) TS3 . Metadata is also available through the Geoscience Australia portal (https://portal.ga.gov.au/metadata/ geochronology-and-isotopes/isotopes/rbsr-isotope-points/ 4cacd9e8-3340-4c27-99fe-48d404e67ca8 TS4 ; last access 22 August 2022)..

## 6   Conclusions

One hundred and twelve catchment outlet sediment samples from the National Geochemical Survey of Australia archive were analysed for strontium (Sr) isotopic composition ($^{87}$Sr/$^{86}$Sr). Geographically, this study covers 529 000 km$^2$ of the Darling–Curnamona–Delamerian (DCD) area in inland southeastern Australia, and is dominantly flat and low-lying with minor bedrock outcrop. The bottom outlet sediment (BOS) samples were retrieved mostly by augering overbank or floodplain landforms near the outlet of large catchments to, on average, 0.6 to 0.8 m depth. Total digestion of milled <2 mm grain-size fractions from these sediments yielded a wide range of $^{87}$Sr/$^{86}$Sr values from a minimum of 0.7089 to a maximum of 0.7511.

A map of the $^{87}$Sr/$^{86}$Sr distribution (isoscape) reveals spatial patterns reflecting the sources and transport pathways of the sediment carried principally down rivers which drain mainly from the east and the north to the west/southwest and, to a lesser degree, by wind which blows dominantly from the west/northwest. Three main source regions of relatively radiogenic ($^{87}$Sr/$^{86}$Sr $\geq\sim$ 0.7270; top quartile) clastic material are recognized. First, the Palaeozoic felsic igneous and sedimentary rocks of the Lachlan geological region to the east and southeast of the DCD area contribute fluvial sediment that is transported west by the Murray River and several tributaries. Second, in the central DCD area, alluvium shed from the topographically proud outcrop of Palaeoproterozoic metamorphic rocks of the Broken Hill geological region disperses radially to the north, west, and southeast. Third, radiogenic material sourced from the Proterozoic to Palaeozoic rocks of the Kanmantoo, Adelaide, Gawler, and Painter geological regions to the west of the area disperses mostly by aeolian processes toward the east and southeast.

These radiogenic sources contrast with the relatively unradiogenic ($^{87}$Sr/$^{86}$Sr $\leq\sim$ 0.7130; bottom quartile) sediments found in a few, mostly internally draining catchments in the central Murray Basin, some Darling Basin catchments in the northeast, and a few Eromanga geological region-influenced catchments in the northwest of the study area.

This study constitutes the largest Sr isoscape in Australia to date and demonstrates the usefulness of $^{87}$Sr/$^{86}$Sr analysis as a tool for understanding geological and geomorphological processes that may impact natural resources management of soil, water, and agriculture. Although we have focused the discussion on sediment provenancing using $^{87}$Sr/$^{86}$Sr data, potential applications could be extended to mineralization, hydrology, food tracing, dust provenancing/sourcing, and historic migrations of people and animals.

**Supplement.** The supplement related to this article is available online at: https://doi.org/10.5194/essd-14-1-2022-supplement.

**Author contributions.** PdC provided the concept, samples, funding, data curation, analysis and visualization, and manuscript writing and editing. AD provided technical guidance, resources and supervision, data curation, and manuscript editing. FD provided technical support and data curation.

**Competing interests.** The contact author has declared that none of the authors has any competing interests.

**Disclaimer.** Patrice de Caritat publishes with the permission from the Chief Executive Officer, Geoscience Australia.

Publisher's note: Copernicus Publications remains neutral with regard to jurisdictional claims in published maps and institutional affiliations.

**Acknowledgements.** The National Geochemical Survey of Australia (NGSA) project would not have been possible without Commonwealth funding through the Onshore Energy Security Program, and Geoscience Australia appropriation (http://www.ga.gov. au/ngsa, last access: 22 August 2022). The strontium isotopic analyses reported here were funded by the Exploring for the Future (EFTF 2020–2024) initiative of the Australian Government. Collaboration with the geoscience agencies of all states and the Northern Territory was essential to the success of the NGSA project, and is gratefully acknowledged. We acknowledge all land owners and custodians, whether private, corporate, and/or traditional, for granting access to the field sites for the purposes of sampling, and Geoscience Australia laboratory staff for assistance with preparing and analysing the samples. We thank Geoscience Australia reviewers Kathryn Waltenberg and David Huston, journal reviewers Patrick De Deckker, Jurian Hoogewerff, and anonymous, and the journal editor for their detailed and constructive critique of our work.

**Financial support.** This research has been supported by the Australian Government (Exploring for the Future).

**Review statement.** This paper was edited by Attila Demény and reviewed by Patrick De Deckker, Jurian Hoogewerff, and one anonymous referee.

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

## Remarks from the language copy-editor

CE1     Please give an explanation of why this needs to be changed. We have to ask the handling editor for approval. Thanks.

## Remarks from the typesetter

TS1     Please confirm URL.

TS2     Please confirm adjustments in the table.

TS3     Please confirm addition of the DOI.

TS4     Please provide a reference list entry including creators, and title.

TS5     Please confirm.

TS6     Thank you for your feedback. I'm afraid this cannot be changed, since the current format is in accordance with our standards. Thank you for your understanding.

TS7     Please check; the DOI should work. Please note that the line numbers will be removed after the proofreading stage.

TS8     Please check; the DOI should work.

TS9     Please check; the DOI should work.

TS10     Please check; the DOI should work.

TS11     Please confirm that the DOI works.

TS12     I apologize, but the DOI does not work with the percent signs. Is this the correct DOI: https://doi.org/10.1016/0375-6742(89)90061-7? If so, please check the title as well with the DOI.

TS13     Please confirm DOI.