# Peer review of "A strontium isoscape of inland southeastern Australia"

_Earth System Science Data, 2022_

## Referee Comment (RC1)

**Review of the paper entitled: A strontium isoscape in inland southeastern Australia**

**by Patrick de Caritat, Anthony Dosseto and Florain Dux**

1. The article contains a very important data set. The use of the data will be of great value for a number of applications such as forensic science to determine the origin of wine for example, the sourcing of airborne dust in marine cores as well as in ice cores in Antarctica, determining the origin and possibly quantity of sediment loads from different catchments like attempted in the paper by Gingele, F.X. & De Deckker, P. (2005). Clay mineral, geochemical and Sr-Nd-isotopic fingerprinting of sediments in the Murray-Darling fluvial system, SE Australia. *Australian Journal of Earth Sciences* 52, 965-974. And finally, it has the potential to determine the extent of migration of early Australians by studying the Sr isotopic composition or human bones.  Unveiled in archaeological sites.

2. This is a significant data set and very thorough. However, no mention is made of the paper by De Deckker **(**2020). Airborne dust traffic from Australia in modern and Late Quaternary times. *Global and Planetary Change* 184, Article 103056, 19 pp. The Sr isotopic data provided in that paper could be included in the data currently being assessed. Hence the data set is not complete as is.

3. The data set is definitely of high quality.

4. The figures are adequate provided they can be enlarged. I am wondering as to whether it may be possible to use a program like google map and click on one site [a cross in the case of the figures] and obtain directly form the data set, the location, description of the site and the raw Sr data? Is it asking too much?

5. By reading the article and downloading the data set, my answer is YES to the question: "would you be able to understand and (re-)use the data set in the future?"

6. **Rating: 1 for excellent**

**Additional comments:**
Line 12: is the word 'coarse' really correct? This is the clay-size fraction (e.g. <2µm).
Line 29: after Madgwicjk et al., 2019) add:  Dust sourcing in ice core' and add a reference or 2 such as Revel-Rolland, M., et al. (2006). Eastern Australia: A possible source of dust in East Antarctica interglacial ice. Earth and Planetary Science 249, 1-13. and De Deckker **(**2020). Airborne dust traffic from Australia in modern and Late Quaternary times. *Global and Planetary Change* 184, Article 103056, 19 pp. but there are many others.
Line 35. Perhaps add the reference of De Deckker, P. (2020) op. cit.
Line 56. Mention also that the fluctuating ENSO signals strongly affects rainfall variability over the years.
Line 62: Really, the Murray River starts much higher near Charlotte Pass with the Snowy River and its mouth is not near Victor Harbour by Goolwa instead! Also after 'Southern Ocean' add in brackets "(Southeast Indian Ocean sector)" to be more precise as the Southern Ocean commences below the Subtropical Convergence.
Line 70 give the range of altitudes form the Flinders Ranges. At present it is a bit misleading as 139 m asl is not representative
Line 77: add: 'plus aeolian dust input"
Line 79: replace 'lack' with 'absence'

Line 220: can you postulate on the origin of the sediments??

Line 235, Perhaps quote some of the values listed in De Deckker (2020) op cit.

Line 245: you should check the following article on Lake Frome east of the Flinders Ranges and quote some of the isotope data from it as it also gives values for groundwaters for the Great Artesian Basin. W. J. Ullman & K. D. Collerson (1994) The Sr-isotope record of Late Quaternary hydrologic changes around Lake Frome, South Australia, Australian Journal of Earth Sciences, 41:1, 37-45, DOI: 10.1080/08120099408728111

Somewhere near line 255: you should mention the possibility of contamination in areas where fertilizers are used as their Sr isotopic composition is well known but can affect some of your results. Fertilisers can be transferred done to eth depths of you sampling. Check Martin CE & McCulloch MT 1999. Nd-Sr isotopic and trace element geochemistry of river sediments and soils in a fertilized catchment, New South Wales, Australia. GCA 63 (2), 287-305.

Line 350 caption. It would be nice in the top figure to list the number of analyses carried out for each boxplot. Concerning the bottom figure, it is a pity in a way to plot a box for each of the single samples for Bancannia and St Vincent as they only represent a single sample.

Line 397: Would you consider reiterating the possible uses of your data such as I mentioned earlier in my comments?

**In summary:** an excellent data set, something that was lacking for Australia. Can the same be done using Nd isotopes as a combination of both Sr and Nd isotopes would a much more formidable tool.

Patrick De Deckker
Research Scholl of Earth Sciences, Australian National University , Canberra.
5 May 2022

---

## Referee Comment (RC2)

**General comments**

Each new Sr or other isotope dataset that is published is a valuable contribution to creating isotope maps of the world and in this case Australia. Collecting samples in the often remote Australian landscape and analysing Sr isotopes are both labour intensive activities and therefore the authors effort and making the data freely available is very much appreciated.

The quality of the analytical work seems excellent as can be expected from a collaboration of an experienced isotope lab with GA, so no concerns there.

**Specific comments**

P1L9: Disagree with "Ultimately....." the way the sentence is written seems to imply that only the locally underlying parent material determines Sr isotope ratios in soil. Later the authors actually argue for other sources also contributing to the Sr signal mix so I suggest rewording this sentence.

P1L15: I question if reporting routine statistic parameters is very relevant for Sr isotope distributions as Sr distributions are not normal, always multimodal (in these kind of mapping surveys).

P1L28: I suggest referring to a good and critical discussion of Sr isotopes for proveniencing by Jane Evans as is nicely discusses its limitations 10.1080/00665983.2021.1911099

P2L30: I note that any reference to the recent work of Bataille on the global Sr map and Hoogewerff on the European Sr map is absent? Either of these publications would provide large datasets to compare the current data with. Also these data sets can be used to show the global or large scale distributions of Sr isotope values, maybe allowing to better underpin the later argument for the suggested bi-modality of the presented data?

P7L170-176, table 1and fig3: What is the evidence for bimodal, does bimodal make sense in the geological context? The CF curve shows several plateaux indicating more than 2 distributions and considering the paucity of measurements in some catchments I would be careful with reading too much in the descriptive stats and making decisions on sub-populations. I note that global data also shows a skewed population with a long RH tail, so how does this data compare with the global data, such comparison might be more relevant that using normal descriptive stats?

P10L225-233: Is there a contradictive argument here? At one place in discussion it is argued that wind-blown deposits are mostly quartz with low Sr isotope values but here it is suggested that radiogenic minerals are blown in?

P13/14 section 4.5 and Fig 6: if some samples are diluted by aeolian quartz and affecting/diluting the 1/Sr ratio, would it be useful to make an additional plot of 87Sr/86Sr versus Rb/Sr or Al/Sr to compensate for the quartz dilution?

P16 fig7B:   I am somewhat concerned about the bias caused by very differencing sample numbers in each category, maybe giving a wrong impression of the distributions?

P17 section 4.6: I suggest exploring the relations with other element a bit more to find out if a regression or machine learning model could predict the Sr isotope data, or maybe this will be attempted in a separate paper?

P17 Conclusions: partly seem a somewhat repeat of stats that have already been mentioned twice, in abstract and discussion.

**Technical corrections**

P2 Fig1: seems small in publication, fig a has a lot of redundant place names?

P4 fig2 Are some colours missing in figure 2A map, most seems white?? And again seems small in publication

P7L174: are 4 digits relevant for summary stats like skewness and kurtosis?

P12 Fig4:  too may digits in $R^2$? what is significance of 3rd and 4th decimals in intercept and/or  $R^2$ if slope has only one digit? Slope would need more significant figures?

P14/15: maybe better to put n=x for each catchment in figure 7a, rather than as text?

P16 Fig 7a: personally, I would prefer a Violin plot as that shows the distributions in each catchment better (when there is enough data)

Jurian Hoogewerff
National Centre Forensic studies, Fac Science & Technology,  University of Canberra.
9 May 2022

---

## Author Comment (AC1)

essd-2022-125-Rebuttal RC1.docx

Title: A strontium isoscape of inland southeastern Australia
Author(s): Patrice de Caritat et al.
MS No.: essd-2022-125
MS type: Data description paper

Dear Editor

Thank you for the opportunity to revise the above manuscript. Our responses to Reviewer 1's comments are below and the revised manuscript file is attached.

1. The article contains a very important data set. The use of the data will be of great value for a number of applications such as forensic science to determine the origin of wine for example, the sourcing of airborne dust in marine cores as well as in ice cores in Antarctica, determining the origin and possibly quantity of sediment loads from different catchments like attempted in the paper by Gingele, F.X. & De Deckker, P. (2005). Clay mineral, geochemical and Sr-Nd-isotopic fingerprinting of sediments in the Murray-Darling fluvial system, SE Australia. *Australian Journal of Earth Sciences* 52, 965-974. And finally, it has the potential to determine the extent of migration of early Australians by studying the Sr isotopic composition or human bones. Unveiled in archaeological sites.

We note the reviewer's assessment of the importance of the contributed dataset and its multiple potential applications, and we thank the reviewer for that.

2. This is a significant data set and very thorough. However, no mention is made of the paper by De Deckker (2020). Airborne dust traffic from Australia in modern and Late Quaternary times. *Global and Planetary Change* 184, Article 103056, 19 pp. The Sr isotopic data provided in that paper could be included in the data currently being assessed. Hence the data set is not complete as is.

The paper mentioned by the reviewer, De Deckker (2020), has been cited and added to the References. As it is not the purpose of this contribution to collate and combine all existing data from the region, we prefer to leave this task to others/later, at this stage. The main purpose of the present ESSD contribution is to present a new dataset.

3. The data set is definitely of high quality.

Thank you.

4. The figures are adequate provided they can be enlarged. I am wondering as to whether it may be possible to use a program like google map and click on one site [a cross in the case of the figures] and obtain directly form the data set, the location, description of the site and the raw Sr data? Is it asking too much?

Full resolution figures will be provided with final submission. In addition, once published, the data will be visible and accessible via the Geoscience Australia portal in exactly the manner suggested.

5. By reading the article and downloading the data set, my answer is YES to the question: "would you be able to understand and (re-)use the data set in the future?"

Thank you.

**6. Rating: 1 for excellent**

Thank you.

Additional comments (line numbers refer to the originally submitted version as reviewed):

Line 12: is the word 'coarse' really correct? This is the clay-size fraction (e.g. <2μm).

L.12: Yes, the term 'coarse' designating the <2 mm fraction is correct, as used in the NGSA project and multiple reports.

Line 29: after Madgwicjk et al., 2019) add: Dust sourcing in ice core' and add a reference or 2 such as Revel-Rolland, M., et al. (2006). Eastern Australia: A possible source of dust in East Antarctica interglacial ice. Earth and Planetary Science 249, 1-13. and De Deckker (2020). Airborne dust traffic from Australia in modern and Late Quaternary times. *Global and Planetary Change* 184, Article 103056, 19 pp. but there are many others.

L.29: Suggested references added.

Line 35. Perhaps add the reference of De Deckker, P. (2020) op. cit.

L.35: We are talking about new projects currently underway at Geoscience Australia here. De Deckker is mentioned now 6 lines above, where relevant.

Line 56. Mention also that the fluctuating ENSO signals strongly affects rainfall variability over the years.

L.56: Sentence 'Long-term weather patterns are strongly affected by El Niño/La Niña cycles.' added.

Line 62: Really, the Murray River starts much higher near Charlotte Pass with the Snowy River and its mouth is not near Victor Harbour by Goolwa instead! Also after 'Southern Ocean' add in brackets "(Southeast Indian Ocean sector)" to be more precise as the Southern Ocean commences below the Subtropical Convergence.

L.62: The source of the Murray River as described ('in the Australian Alps at 1430 m above sea level (asl) (on the border between NSW and Vic)') is entirely in agreement with the reviewer's comment. The mouth of the river is described in terms of the maps shown, where the closest labelled town is Victor Harbour. Of course at higher resolution, Goolwa would be visible. The detail about the Southern Ocean label seems unnecessarily detailed in the context; this map/label is for general orientation.

Line 70 give the range of altitudes form the Flinders Ranges. At present it is a bit misleading as 139 m asl is not representative

L.70: The mean altitude of the study area is 139 m asl. It is mentioned that both the Victorian Highlands and the Flinders Ranges have elevations >900 m asl.

Line 77: add: 'plus aeolian dust input"

L.77: Mention of aeolian processes has been added.

**Line 79: replace 'lack' with 'absence'**

L.79: Done.

**Line 220: can you postulate on the origin of the sediments??**

L.220: We wouldn't be able to speculate further than stated, given the spatial resolution of the samples.

**Line 235, Perhaps quote some of the values listed in De Deckker (2020) op cit.**

L.235: As that study is concerned with a much finer grainsize fraction than the dataset presented, we feel it is unwarranted to quote numbers that may not be comparable. This could be part of a discussion in a separate study collating all regional Sr isotope data.

**Line 245: you should check the following article on Lake Frome east of the Flinders Ranges and quote some of the isotope data from it as it also gives values for groundwaters for the Great Artesian Basin. W. J. Ullman & K. D. Collerson (1994) The Sr-isotope record of Late Quaternary hydrologic changes around Lake Frome, South Australia, Australian Journal of Earth Sciences, 41:1, 37-45, DOI: 10.1080/08120099408728111**

L.245: Thank you for the suggestion. The reference to Ullman and Collerson has been added in Section 4.3, however, where it seems more appropriate (as the values reported are relatively unradiogenic compared to the dataset presented here).

**Somewhere near line 255: you should mention the possibility of contamination in areas where fertilizers are used as their Sr isotopic composition is well known but can affect some of your results. Fertilisers can be transferred done to eth depths of you sampling. Check Martin CE & McCulloch MT 1999. Nd-Sr isotopic and trace element geochemistry of river sediments and soils in a fertilized catchment, New South Wales, Australia. GCA 63 (2), 287-305.**

Somewhere near L.255: Thank you for the suggestion. The reference to Martin and McCulloch and a discussion of the potential role of fertilizers have been added to Subsection 4.3.

**Line 350 caption. It would be nice in the top figure to list the number of analyses carried out for each boxplot. Concerning the bottom figure, it is a pity in a way to plot a box for each of the single samples for Bancannia and St Vincent as they only represent a single sample.**

L.350: Thank you for the suggestion. The caption of Figure 7a has been modified to include the number of observations for each boxplot, as suggested. As for Figure 7b we believe it is best to show the values reported, as this is more informative than not showing them at all. It is clear from the size of the box in the histogram that there are single data points, thus this is not misleading. In fact it could spur further data collection in these regions.

**Line 397: Would you consider reiterating the possible uses of your data such as I mentioned earlier in my comments?**

L.397: Thank you for the suggestion. We have taken this advice.

**In summary:** an excellent data set, something that was lacking for Australia. Can the same be done using Nd isotopes as a combination of both Sr and Nd isotopes would a much more formidable tool.

We wish to thank reviewer #1 for devoting time to this task. The comments are much appreciated.

The collection of Nd isotope data on NGSA samples would indeed deliver a formidable tool and has actually been piloted; more work could be done if/when funding is identified.

---

## Author Comment (AC2)

essd-2022-125-Rebuttal RC2.docx

Title: A strontium isoscape of inland southeastern Australia
Author(s): Patrice de Caritat et al.
MS No.: essd-2022-125
MS type: Data description paper

Dear Editor

Thank you for the opportunity to revise the above manuscript. Our responses to Reviewer 2's comments are below and the revised manuscript file is attached.

General comments

**General comments**

Each new Sr or other isotope dataset that is published is a valuable contribution to creating isotope maps of the world and in this case Australia. Collecting samples in the often remote Australian landscape and analysing Sr isotopes are both labour intensive activities and therefore the authors effort and making the data freely available is very much appreciated.

The quality of the analytical work seems excellent as can be expected from a collaboration of an experienced isotope lab with GA, so no concerns there.

Thank you for the comments.

Specific comments

P1L9: Disagree with "Ultimately….." the way the sentence is written seems to imply that only the locally underlying parent material determines Sr isotope ratios in soil. Later the authors actually argue for other sources also contributing to the Sr signal mix so I suggest rewording this sentence.

L.9: This was an overly broad generalisation, so thank you for pointing that out. We have modified the sentence to clarify the case of different components, in-situ and allochthonous.

P1L15: I question if reporting routine statistic parameters is very relevant for Sr isotope distributions as Sr distributions are not normal, always multimodal (in these kind of mapping surveys).

L.15: Although we understand the remark, there is nothing fundamentally wrong with reporting univariate statistics for any population (especially when plots are shown to illustrate the distributions). It is deplorable that this is rarely done these days in our view. We see it as part of a basic description of the data.

P1L28: I suggest referring to a good and critical discussion of Sr isotopes for proveniencing by Jane Evans as is nicely discusses its limitations 10.1080/00665983.2021.1911099

L.28: Reference to Madgwick et al. (2021) has been added.

P2L30: I note that any reference to the recent work of Bataille on the global Sr map and Hoogewerff on the European Sr map is absent? Either of these publications would provide large datasets to compare the current data with. Also these data sets can be used to show the global or large scale distributions of Sr isotope values, maybe allowing to better underpin the later argument for the suggested bi-modality of the presented data?

L.30: References to Bataille and Hoogewerff have been added.

P7L170-176, table 1and fig3: What is the evidence for bimodal, does bimodal make sense in the geological context? The CF curve shows several plateaux indicating more than 2 distributions and considering the paucity of measurements in some catchments I would be careful with reading too much in the descriptive stats and making decisions on sub-populations. I note that global data also shows a skewed population with a long RH tail, so how does this data compare with the global data, such comparison might be more relevant that using normal descriptive stats?

L.170-176: See comment above about descriptive statistics. We agree that the distribution is probably more complex, in detail, than just bimodal. We have qualified our words accordingly and added a comparison to the world dataset.

P10L225-233: Is there a contradictive argument here? At one place in discussion it is argued that wind-blown deposits are mostly quartz with low Sr isotope values but here it is suggested that radiogenic minerals are blown in?

L.225-233: This is not contradictory in our view. The Sr isotopic composition of any aeolian contribution will depend on where the transported material is sourced from and its nature (e.g., mineralogy). Most often it will be quartzose silt and fine sand with a low Sr isotopic signature, but if a nearby source region has radiogenic minerals and they are eroded and available for winnowing, they can be transported and deposited down-wind. This is what we speculate here. In other words it's not the process that control the Sr isotopic value of the material, but the material itself and hence its source(s).

P13/14 section 4.5 and Fig 6: if some samples are diluted by aeolian quartz and affecting/diluting the 1/Sr ratio, would it be useful to make an additional plot of 87Sr/86Sr versus Rb/Sr or Al/Sr to compensate for the quartz dilution?

Fig.6: This is an excellent suggestion and we had done this. The Si/Sr plot is below, but does not add anything particularly useful, in this case.

[Figure]

YX Plot [Sri_map, Si/Sr]

**Si/Sr : Sri_map**

Legend
Combined (11)
**REGNAME**
- Adelaide Region
- Arrowie Region
- Bancannia Region
- Broken Hill Region
- Darling Region
- Eromanga Region
- Kanmantoo Region
- Lachlan Region
- Murray Region
- St Vincent Region
- Wonominta Region

**P16 fig7B:** I am somewhat concerned about the bias caused by very differencing sample numbers in each category, maybe giving a wrong impression of the distributions?

Fig.7b: This comment mirrors that of reviewer #1 and has been addressed by explicitly giving the size of each subpopulation in the caption. As we expand our work on Sr isoscapes in Australia, more geological regions will become increasingly populated with Sr isotopic data so we hope to alleviate this in the future.

**P17 section 4.6:** I suggest exploring the relations with other element a bit more to find out if a regression or machine learning model could predict the Sr isotope data, or maybe this will be attempted in a separate paper?

Subsection 4.6: This is beyond the scope of this paper, which is focused on presenting Australia's largest Sr isotopic dataset to date. As we and other groups publish more Sr isotopic data, the possibility of machine learning/modelling 87Sr/86Sr distributions more widely and with higher resolution will be explored in detail.

**P17 Conclusions:** partly seem a somewhat repeat of stats that have already been mentioned twice, in abstract and discussion.

Conclusions: Yes. In our experience, many people read the Conclusions of papers without having time to read the whole article. Thus we take the view that it is useful to restate the fundamentals of the contribution, at the risk of a degree of repetition for the assiduous reader (*cfr* reviewer 1's comment to reiterate the applications of Sr isotopes in the Conclusions). Being a new Sr isotopic dataset for Australia, and a significant one on terms of size, it is particularly appropriate to have basic statistics in the Conclusions. We think the Conclusions are a detailed and informative yet concise and correct summary of our contribution.

Technical corrections

**P2 Fig1: seems small in publication, fig a has a lot of redundant place names?**

Fig.1: Full resolution figures will be provided for final publication; they can be zoomed in for detailed analysis. Named places are based on a population threshold. Ultimately, the interested user will be able to view and manipulate the dataset on the GA portal so all presentation choices can be made on the fly.

**P4 fig2 Are some colours missing in figure 2A map, most seems white?? And again seems small in publication**

Fig.2: Yes the maps are correct. Most of the region is covered by the Regolith/Other category as discussed. See comment above about resolution and future portal availability.

**P7L174: are 4 digits relevant for summary stats like skewness and kurtosis?**

L.174: We used the same number of decimal places as for the Sr isotopic data, the choice of which is justified in the text (Subsection 3.3).

**P12 Fig4: too may digits in $R^2$? what is significance of 3$^{rd}$ and 4$^{th}$ decimals in intercept and/or $R^2$ if slope has only one digit? Slope would need more significant figures?**

Fig.4: The slopes are in scientific notation because smaller than the four decimal places would be useful for. It is clear the slopes of the two regression lines are similar as they are almost perfectly parallel. The R values have now been rounded to two decimal places in the figure and text.

**P14/15: maybe better to put n=x for each catchment in figure 7a, rather than as text?**

Fig.7: N per region (not catchment) has been added as requested by reviewer #1.

**P16 Fig 7a: personally, I would prefer a Violin plot as that shows the distributions in each catchment better (when there is enough data)**

Fig.7a: Agreed, violin plots would be more informative than standard box plots, but these are not available in the software used.

We wish to thank reviewer #2 for devoting time to this task. The comments are much appreciated.

---

## Author Comment (AC3)

essd-2022-125-Rebuttal RC3.docx

Title: A strontium isoscape of inland southeastern Australia
Author(s): Patrice de Caritat et al.
MS No.: essd-2022-125
MS type: Data description paper

Dear Editor

Thank you for the opportunity to revise the above manuscript. Our responses to Reviewer 3's comments are below and the revised manuscript file is attached.

**General comments**

The paper of de Caritat et al. presents a novel robust database of non-biovailable Sr isotope ratios from southeastern Australia. In general I think that the measured data are of high-quality and well-constrained in the framework of Australia geology. However, I'm not convinced that these data can be used to trace the provenance of biological samples (i.e. isoscape purpose). Indeed, as the authors themselves wrote, non-bioavailable Sr isotopes are rarely good proxies for biological materials, due to the different end-members contributing to the two classes (i.e. bioavailable vs. non-bioavailable) final isotope ratios. This clearly limits the possible use of such data. Moreover, the fact that the samples represent 'averaged' catchment site (5200 km$^2$ on average) is definitely 'blurring' the potential resolution and thus prediction power of any isoscape built on the data. The authors clearly state this on L200 page 8, however I think that such issues need to be discussed more in depth maybe in the introduction, to warn the reader on the limits of the current dataset. My suggestion is also to adapt the title as: 'A non-bioavailable Sr isoscape of inland southeastern Australia'.

Thank you for the comments. We agree that our total Sr data will not be directly useable by researchers interested in bioavailable Sr patterns and processes. Nor have we proposed so in the paper. Our focus has been on geological processes that involve whole minerals such as tracing fluvial sediment and aeolian dust provenance and identifying major sources/reservoirs in the geology. Thus our methodology (from sample selection to analytical method) was designed with this aim in mind. However, total Sr isotope will be a useful predictor, we expect, of bioavailable Sr, e.g., via machine learning (as mentioned in Subsection 4.1). In fact we plan to develop some new research in that area shortly.

Sampling density is always a trade-off between detail and coverage. As we are interested in the first instance in large-scale processes, an ultralow density sampling scheme such as afforded by the NGSA samples seems appropriate. Of course, it would be desirable to fill-in the sampling grid with smaller catchment data, which future studies may well take on. We have added some more details in the Introduction, as suggested, but not in Section 1 rather in Subsection 3.1, Materials:

*The sampling medium and density were both strategically chosen in the NGSA project to prioritise coverage over resolution. This was justified by the fact that the NGSA was Australia's first, and to date only, fully integrated, internally consistent geochemical survey with a truly national scope. In terms of the DCD, it is clear that these choices have implications on the granularity of the patterns revealed by the Sr isoscape; as the collection of Sr isotope data in Australia using NGSA samples grows in the future, it is hoped the value of coverage will prevail over a relative low resolution of detailed features.*

We acknowledge the suggestion for a modification of the title of the paper, but respectfully disagree (see below).

**Specific comments**

Isoscape terminology. Although many works on isoscape are purely descriptive, an isoscape should represent a modelled map of a specific isotope distribution. This mean that the data should be accompanied by a modelling outline and validation, to show the prediction power of the model itself. This is the main reason why I feel that this paper mostly represents a new dataset rather than an isoscape of the area. I'm not asking to entirely change the terminology used in the manuscript, but I think it is something that we (as community) should keep in mind for future works.

Isoscape terminology: Although it is possible that some researchers exclusively associate the term isoscape with bioavailability, this is compatible with neither the original meaning of the

term nor its broader usage (e.g., for other isotopic systems or for non biological media such as groundwater or precipitation). The earliest reference to isoscape we could find cites a personal communication from G.J. Bowen to the author, Keith Hobson, mentioning *'"isotopic landscape" or "isoscape"'* (Hobson, 2005). This is also how West et al. (2010) in their authoritative book entitled **Isoscapes** define the word: *'This volume provides a comprehensive overview of the theory, methods, and applications that are enabling new disciplinary and cross-disciplinary advances through the study of "isoscapes": isotopic landscapes.'* Thus we respectfully maintain that we present \*a\* strontium isoscape of inland SE Australia (noting that we do not state \*the\* strontium isoscape). We agree that all in the community should keep this difference in usage in mind for future works.

Technical corrections

'Robust standard deviation' is a truly un-used descriptive statistic term. I suggest to report data as median ± median absolute deviation (MAD).

Robust standard deviation: we have changed this to MAD. Thank you for pointing that out.

Table 1: a single line table is not very useful. Maybe you can add here the descriptive statistics for each geological region (as Figure 7).

Thank you for the excellent suggestion, which we have taken on board. Table 1 now has the various regions listed.

L172 p.7: the link is incomplete. Will It be updated during/after the publication process?

Yes, the link will be updated at the proofs stage. When the paper is accepted, we will make the dataset 'live' on the portal.

We wish to thank reviewer #3 for devoting time to this task. The comments are much appreciated.